**REPORT**

# Phosphoregulation of tropomyosin is crucial for actin cable turnover and division site placement

Saravanan Palani[1], Darius V. Köster[1], Tomoyuki Hatano[1], Anton Kamnev[1], Taishi Kanamaru[1], Holly R. Brooker[3], Juan Ramon Hernandez-Fernaud[2], Alexandra M.E. Jones[2], Jonathan B.A. Millar[1], Daniel P. Mulvihill[3], and Mohan K. Balasubramanian[1]

**Tropomyosin is a coiled-coil actin binding protein key to the stability of actin filaments. In muscle cells, tropomyosin is subject to calcium regulation, but its regulation in nonmuscle cells is not understood. Here, we provide evidence that the fission yeast tropomyosin, Cdc8, is regulated by phosphorylation of a serine residue. Failure of phosphorylation leads to an increased number and stability of actin cables and causes misplacement of the division site in certain genetic backgrounds. Phosphorylation of Cdc8 weakens its interaction with actin filaments. Furthermore, we show through in vitro reconstitution that phosphorylation-mediated release of Cdc8 from actin filaments facilitates access of the actin-severing protein Adf1 and subsequent filament disassembly. These studies establish that phosphorylation may be a key mode of regulation of nonmuscle tropomyosins, which in fission yeast controls actin filament stability and division site placement.**

## Introduction

Tropomyosins are dimeric coiled-coil proteins that interact with formin nucleated actin filaments (Longley, 1975; Gunning et al., 2015; Khaitlina, 2015). Tropomyosins have been studied extensively in the context of muscle contraction, in which they associate with F-actin in thin filaments and prevent or promote interaction of myosin II, present in thick filaments, with the cognate binding sites on actin, in thin filaments, in a calcium-dependent manner (Spudich and Watt, 1971; Gergely, 1974; Szent-Györgyi, 1975; Chalovich et al., 1981; Perry, 2001; Brown and Cohen, 2005; Wakabayashi, 2015). Tropomyosins are also present in nonmuscle cells, where they play important functions in actin filament stability, cell polarity, and cytokinesis (Liu and Bretscher, 1989; Balasubramanian et al., 1992; Perry, 2001; Gunning et al., 2015; Khaitlina, 2015). How tropomyosins are regulated in nonmuscle cells remains unknown, since troponins are not expressed in these cells.

*Schizosaccharomyces pombe* is an attractive organism for the study of the actin cytoskeleton and its role in cell function (Pollard and Wu, 2010; Cheffings et al., 2016). Division of *S. pombe* involves an actomyosin ring, which is positioned in the cell middle through a stimulatory pathway involving the anillin-like protein Mid1 and an inhibitory pathway, involving the proteins Tea1, Tea4, and Pom1 (Fankhauser et al., 1995; Chang et al., 1996; Celton-Morizur et al., 2006; Padte et al., 2006; Huang et al., 2007). Fission yeast expresses a single tropomyosin encoded by the *cdc8* gene (Balasubramanian et al., 1992). Cdc8-tropomyosin is a component of interphase actin cables and the cytokinetic actomyosin ring (Balasubramanian et al., 1992; Skau and Kovar, 2010). Cdc8 binding has been shown to protect F-actin severing by Adf1/Cofilin (Skau and Kovar, 2010; Christensen et al., 2017), but how Cdc8 binding to actin filaments is modulated remains unknown.

## Results and discussion

### Cdc8 phosphorylation controls actin dynamics during interphase

Previous work (Kettenbach et al., 2015; Swaffer et al., 2018) has found that Cdc8 is phosphorylated on multiple residues, raising the possibility that tropomyosin is regulated by phosphorylation. Independently, we purified Cdc8 as a heat-stable polypeptide from *S. pombe* lysates, which was phosphorylated on a number of residues, one of which, S125, has been found to be phosphorylated in our work (Fig. 1, A and B) and in previously published work (Kettenbach et al., 2015; Swaffer et al., 2018), which we characterize herein. Tropomyosins are composed of repeating heptad units in which positions a and d contain hydrophobic amino acids and positions b, c, e, f, and g are occupied by charged amino acids (Brown et al., 2001; Barua et al., 2011). Serine 125 occupies position f in the heptad repeat found in Cdc8 and was therefore expected to be on the surface of the protein (Fig. 1 C). To understand the cellular function of Cdc8 S125 phosphorylation, we generated yeast strains in which WT *cdc8*, *cdc8*-S125A (to mimic a dephosphorylated state), and *cdc8*-S125E

[1]Centre for Mechanochemical Cell Biology and Division of Biomedical Sciences, Warwick Medical School, University of Warwick, Coventry, UK; [2]School of Life Sciences, University of Warwick, Coventry, UK; [3]School of Biosciences, University of Kent, Canterbury, Kent, UK.

Correspondence to Mohan K. Balasubramanian: M.K.Balasubramanian@warwick.ac.uk; Saravanan Palani: s.palani@warwick.ac.uk.

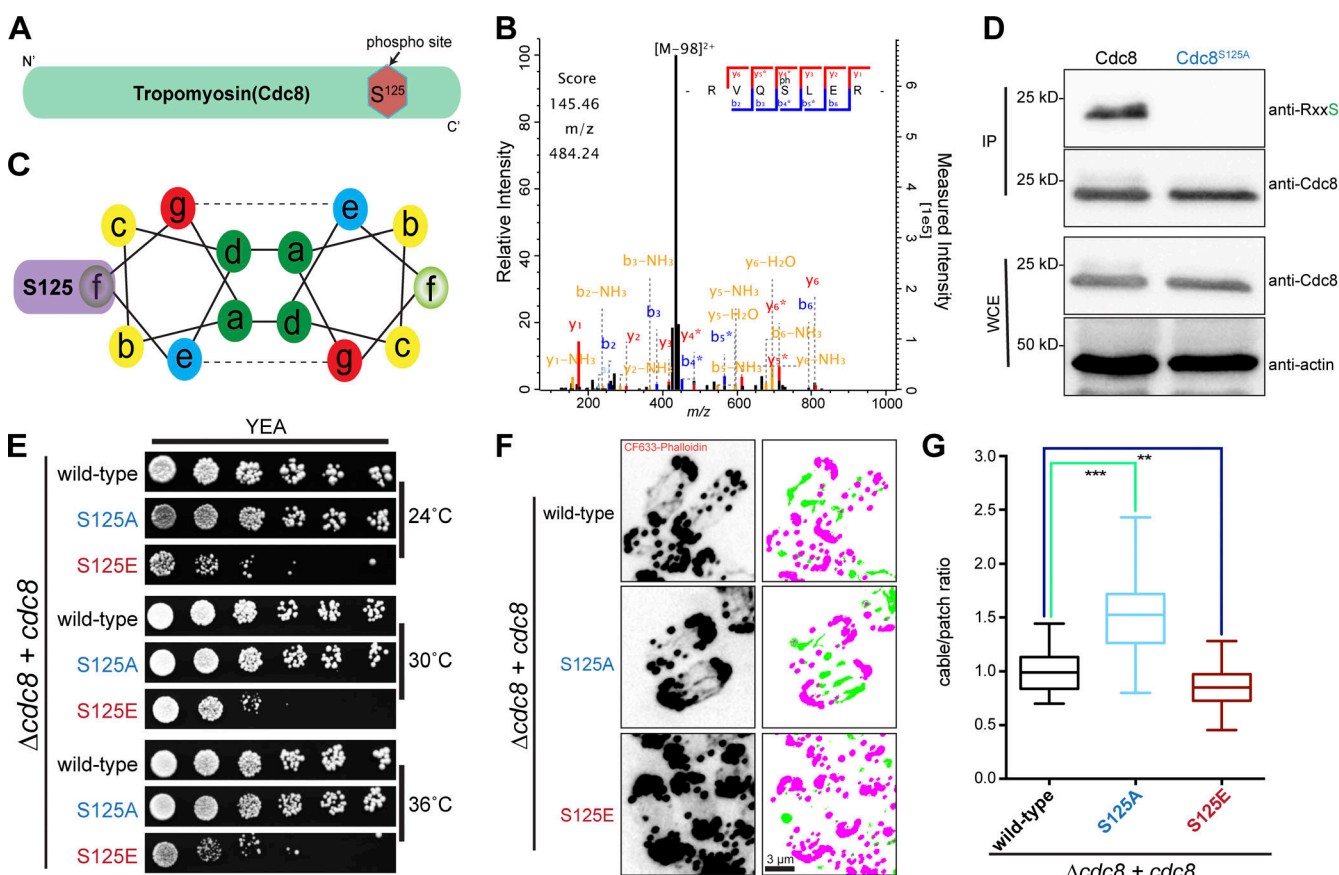

Figure 1. **Phosphoregulation of Cdc8-tropomyosin is important for actin cable stability and cytokinesis. (A)** Schematic representation of the phosphorylation site on tropomyosin cdc8. **(B)** Graphical representation of the coiled-coil heptad repeat organization of Cdc8. Serine (S125) residue is positioned at f on the outside of the coiled-coil heptad. **(C)** Liquid chromatography-MS/MS spectra of one representative phosphopeptide carrying S125P. **(D)** Recognition of Cdc8 S125P, but not Cdc8 S125A, by an antibody against RXXSP. Cell lysates from *cdc8* and *cdc8*-S125A cells were immunoprecipitated (IP) with Cdc8 antibodies and immune complexes or cell lysates (whole-cell extract; WCE) were immunoblotted using antibodies against Cdc8, RxxSp, and actin (loading control). **(E)** Assessment of colony formation by *cdc8* mutants. 10-fold serial dilutions of *cdc8+*, *cdc8*-S125A, and *cdc8*-S125E inoculated on YES plates were incubated at indicated temperatures for 3 d. **(F)** Exponentially growing *cdc8+*, *cdc8*-S125A, and *cdc8*-S125E at 24°C were stained for actin structures with CF-633-phalloidin. Also shown are the corresponding segmentation of filaments (green) and patches (pink). Scale bar represents 3 µm. **(G)** Quantification of F by calculating the ratio of filament to patch area normalized to the average found in WT cells; n = [19, 30, 31] fields of view; box depicts median and 25% to 75% range, whiskers depict maximum and minimum. ***, P < 0.0001; **, P < 0.008.

(to mimic a constitutively phosphorylated state) genes were expressed from the *lys1* locus under control of the native *cdc8* promoter in a *cdc8*-null mutant. Phosphorylation of the ERRVQS125L heptad could be recognized by a commercially available RXXSP-specific antibody when tested on Cdc8 purified from WT cells, but not from *cdc8*-S125A cells (Fig. 1 D).

We noted that the WT and *cdc8*-S125A mutant were nearly indistinguishable in colony formation at 24°C, 30°C, and 36°C (Fig. 1 E). By contrast, *cdc8*-S125E was slow to form colonies at these temperatures (Fig. 1 E). By staining with fluorescent-phalloidin (CF-633), we observed more actin cables in *cdc8*-S125A than in control cells (Fig. 1, F and G). Conversely, actin cables were shorter and less abundant in *cdc8*-S125E (Fig. 1 F). Furthermore, staining of WT, *cdc8*-S125A, and *cdc8*-S125E with antibodies against Cdc8 reinforced the view that actin cables are more prevalent in *cdc8*-S125A compared with WT cells, and they were undetectable in *cdc8*-S125E (Fig. S1). These experiments indicate that actin cable length and stability are controlled by Cdc8 phosphorylation at S125.

**Aberrant Cdc8 phosphorylation causes defects in cytokinesis**

We noted that a large fraction of *cdc8*-S125E, unlike WT, displayed multiple nuclei with abnormal septa (Fig. 2, A and B). A temperature-dependent low-penetrance septation defect was also observed in the *cdc8*-S125A mutant (split septa, Fig. 2, A and B), suggesting that the phosphoregulation of tropomyosin is required for normal cytokinesis. To analyze this further, we performed time-lapse imaging to quantitatively assess the dynamics of actomyosin ring assembly and contraction in *cdc8*-S125A and *cdc8*-S125E mutants, using Rlc1-3GFP and Atb2-mCherry as markers of the actomyosin ring and the mitotic spindle. Actomyosin ring assembly and contraction kinetics were similar in *cdc8* and *cdc8*-S125A, while *cdc8*-S125E had abnormal actomyosin rings whose assembly took significantly longer than in WT (Fig. 2 C and D; WT, 19.6 ± 1.3 min; *cdc8*-S125E, 42.7 ± 5.8 min). In addition, actomyosin ring contraction was aberrant and took significantly longer in *cdc8*-S125E (53 ± 17 min) compared with WT (31 ± 2.7 min; Fig. 2 D). Collectively, these experiments suggest that persistent

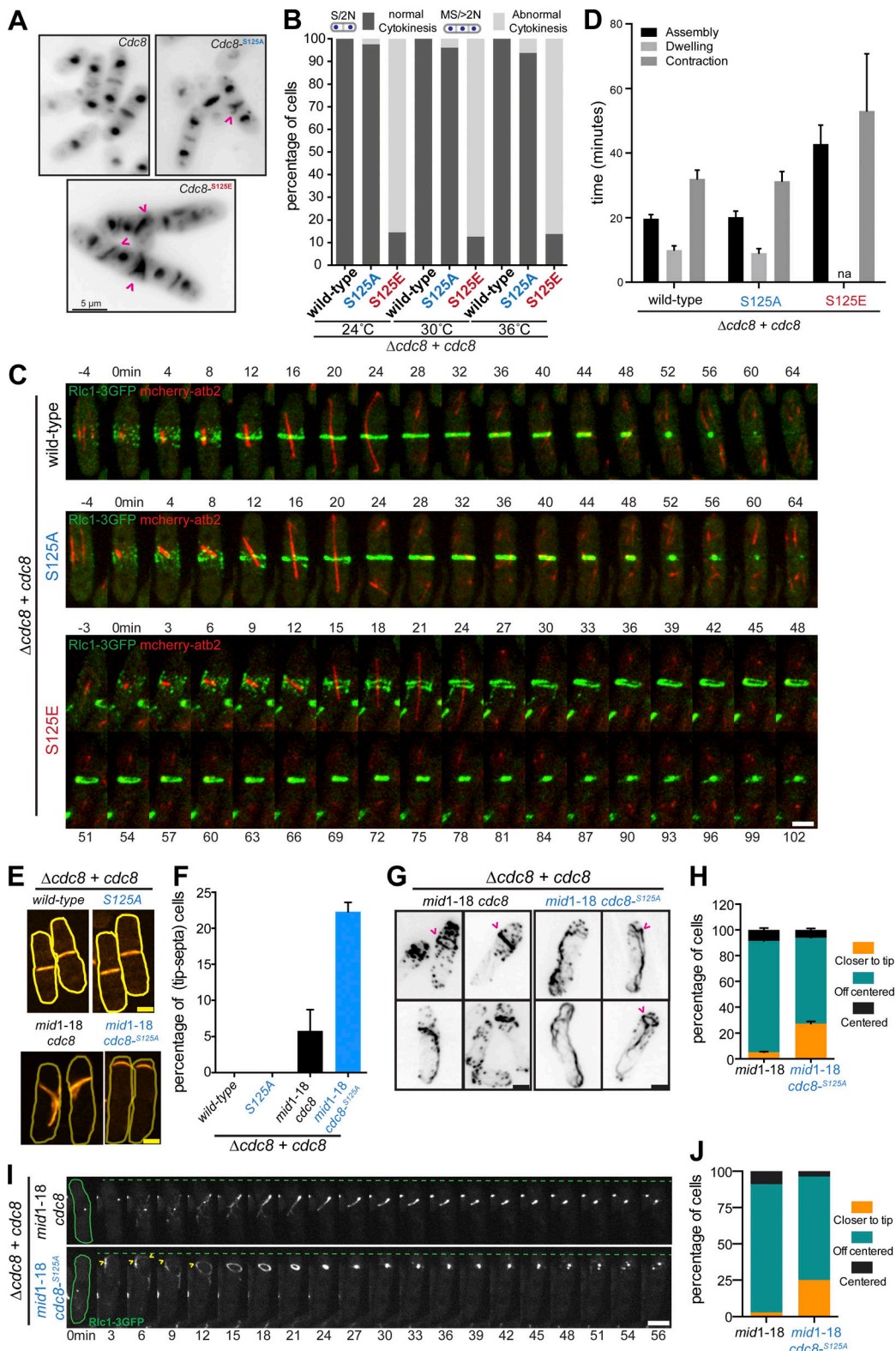

Figure 2. **A phosphomimetic Cdc8 mutant is affected for actin cable stability, actomyosin ring assembly, and contraction. (A)** Exponentially growing *cdc8*+, *cdc8*-S125A, and *cdc8*-S125E at 24°C were fixed with 4% PFA. DAPI and anillin blue staining were used to visualize the DNA and septum. Abnormal septa in *cdc8*-S125E mutants (∼85% from >200 cells examined in two independent experiments) and split septa in cdc8-S125A (∼3% from two independent experiments; 200 cells with septa examined in each case) are highlighted with pink arrows. Scale bar represents 5 μm. **(B)** Quantification of cytokinetic defects in A. *cdc8*, *cdc8*-S125A, and *cdc8*-S125E cells following staining with DAPI (nucleus) and anillin blue (septa; *n* >300 each). **(C)** Time-lapse series of log-phase cells of the indicated genotypes (*cdc8* [*n* = 23], *cdc8*-S125A [*n* = 36], and *cdc8*-S125E [*n* = 24]) expressing 3GFP-tagged myosin regulatory light chain (*rlc1*-3GFP) and mCherry-tagged tubulin (mCherry-*atb2*). Cells were grown at 24°C and imaged at 24°C. Images shown are maximum-intensity projections of z-stacks. Scale bar

**Palani et al.**
Phosphoregulation of tropomyosin in actin function

**Journal of Cell Biology**

3550

represents 3 µm. **(D)** Quantification of C. Timings of ring assembly, maturation, and contraction are shown. na, not applicable. **(E)** Exponentially growing WT, *cdc8*-S125A, *mid1*-18 *cdc8*, and *mid1*-18 *cdc8*-S125A were shifted from 24°C to 36°C for 3 h and stained with anillin blue. Scale bar represents 3 µm. **(F)** Quantification of septum position in E. *n* > 120 cells each, from two independent experiments. **(G)** Exponentially growing *mid1*-18 and *mid1*-18 *cdc8*-S125A were shifted from 24°C to 36°C for 3 h, fixed, and stained for actin structures with CF-633-phalloidin. Scale bar represents 3 µm. **(H)** Quantification of the actin ring position in G (*n* > 150 cells each, from two independent experiments). **(I)** Time-lapse series of exponentially growing *mid1*-18 [*n* = 34] and *mid1*-18 *cdc8*-S125A [*n* = 28] cells expressing (*rlc1*-3GFP). Cells were grown at 24°C and shifted to 36°C for 3 h before imaging. Images shown are maximum-intensity projections of z-stacks. Scale bar represents 3 µm. **(J)** Quantification of the actomyosin ring position in I (*mid1*-18, *n* = 34; *mid1*-18 *cdc8*-S125A *n* = 28). Error bars represent SD.

phosphorylation of Cdc8 causes defects in actomyosin ring assembly and contraction.

## Cdc8 phosphorylation is required for correct division site placement

Previous work has shown that division site placement in fission yeast is achieved by two overlapping mechanisms (Huang et al., 2007), requiring Mid1 (Chang et al., 1996; Sohrmann et al., 1996; Wu et al., 2003) and Pom1-kinase (Bähler and Pringle, 1998). To assess whether Cdc8 was a relevant substrate for Pom1 in modulating tip occlusion, we compared the position of the division septum in WT, *cdc8*-S125A, *mid1*-18, and *mid1*-18 *cdc8*-S125A (Fig. 2 E). WT and *cdc8*-S125A divided medially (Fig. 2, E and F; and Fig. S2, A–C). As previously observed (Huang et al., 2007), *mid1*-18 divided at nonmedial locations, but only 5.8 ± 2.9% of cells displayed division septa at the extreme ends. By contrast, 22.3 ± 1.3% of *mid1*-18 *cdc8* S125A had extreme end division septa (Fig. 2, E and F; and Fig. S2, D–F). These experiments suggested that failure of Cdc8 phosphorylation on S125 significantly abrogates tip occlusion of division septa.

To understand the molecular basis of abrogation of tip occlusion in *mid1*-18 *cdc8*-S125A, we investigated the organization of the actomyosin ring using CF633-phalloidin staining (Fig. 2, G and H) and, through live imaging, using Rlc1-3GFP as a ring marker. We found that F-actin and Rlc1-3GFP invaded the regions closest to the cell tip region in ∼27% of *mid1*-18 *cdc8*-S125A, but only in <5% of *mid1*-18 (Fig. 2, H–J). These findings suggest that actin filament destabilization near the cell ends prevents actomyosin ring formation and septation at cell ends.

## Cdc8 phosphorylation by Pom1 reduces its affinity for actin filaments

To understand the biochemical basis by which tropomyosin phosphorylation influences actin filament stability, we purified amino-terminally acetylated WT and mutant Cdc8 proteins from bacteria (referred to as ^AceCdc8, ^AceCdc8-S125A, and ^AceCdc8-S125E). Circular dichroism (CD) revealed no significant difference in structure and thermostability of each protein (Fig. S3, A–D). Immunoblotting of *S. pombe* cell lysates showed that these proteins were similarly stable at all temperatures (Fig. S3 E), thereby ruling out structure and stability of the mutants as factors in the observed phenotypes.

To investigate whether phosphorylation of tropomyosin influences its binding to F-actin, we performed centrifugation-assisted sedimentation assays of F-actin with an acetyl-mimicking version of Cdc8 (referred to as ASCdc8) or WT or mutant Cdc8 at different molar ratios. We used the acetyl-mimicking version of Cdc8 in this and subsequent work due to the ease of its production as well as to be consistent with other

work in this field (Christensen et al., 2017). We found that ASCdc8-S125E binds to F-actin with a much lower affinity (Fig. 3, A and B). To monitor dynamic interaction of ASCdc8 with F-actin at a molecular level, we made fluorescently labeled actin and ASCdc8 mutants and examined their interaction using total internal reflection fluorescence (TIRF) microscopy, as previously described (Christensen et al., 2017). To fluorescently label Cdc8 with thiol-reactive dyes, we introduced a nonnative cysteine at the location I76C and checked that this did not affect the protein's properties (Fig. S3, F and G). In agreement with the sedimentation assay, we found that Cdc8 decorated F-actin in a cooperative manner, reaching 50% decoration at 0.4 µM, whereas ASCdc8-S125E association with actin was ≥10 times less efficient (Fig. 3, C and D). However, ASCdc8 and ASCdc8-S125E bound actin similarly, at a saturating concentration of 8 µM (Fig. S3 H).

Previous work has shown that S125 on Cdc8 was identified in a large-scale phosphoproteomics screen designed to identify DYRK-kinase Pom1 substrates (Kettenbach et al., 2015). We independently confirmed Pom1-mediated phosphorylation of Cdc8 using three strategies. terially produced WT Pom1 phosphorylated Cdc8 (but not Cdc8S125A and a kinase-dead form of Pom1) failed to phosphorylate Cdc8. Second, the amount of phospho-Cdc8 was reduced in *pom1Δ* (∼30% reduction compared with WT levels) and *pom1AS* (∼50% reduction compared with WT levels) mutants (Fig. S3, I and J). Third, the addition of Pom1 and ATP resulted in rapid dissociation of ASCdc8 from F-actin (Fig. 3, E and F), consistent with Pom1 phosphorylation of Cdc8. No effect was observed when catalytically inactive (kinase-dead) Pom1 mutant (Pom1-KD) and ATP were added to the Cdc8–F-actin complex, providing support to the idea that Pom1-dependent Cdc8 phosphorylation abrogates its interaction with F-actin (Fig. 3, E and F; and Video 1). To establish that the key Pom1 phosphorylation site on Cdc8 is S125, we conducted the same experiments with ASCdc8-S125A and found that neither Pom1-WT nor Pom1-KD altered F-actin association of ASCdc8-S125A (Fig. 3, G and H).

## Cdc8 phosphorylation enables actin filament severing by Adf1/cofilin

Previous work has shown that Cdc8 and the actin-severing protein Adf1 compete for actin association (Christensen et al., 2017). We hypothesized that the weakened interaction between Cdc8 and F-actin might facilitate Adf1-mediated actin severing. Indeed, incubation of ASCdc8-decorated F-actin with Adf1 resulted in only a slight reduction in the average F-actin length, whereas F-actin decorated with ASCdc8-S125E (at a 10-fold higher concentration to achieve stable decoration) displayed a significant reduction in F-actin length upon addition of Adf1

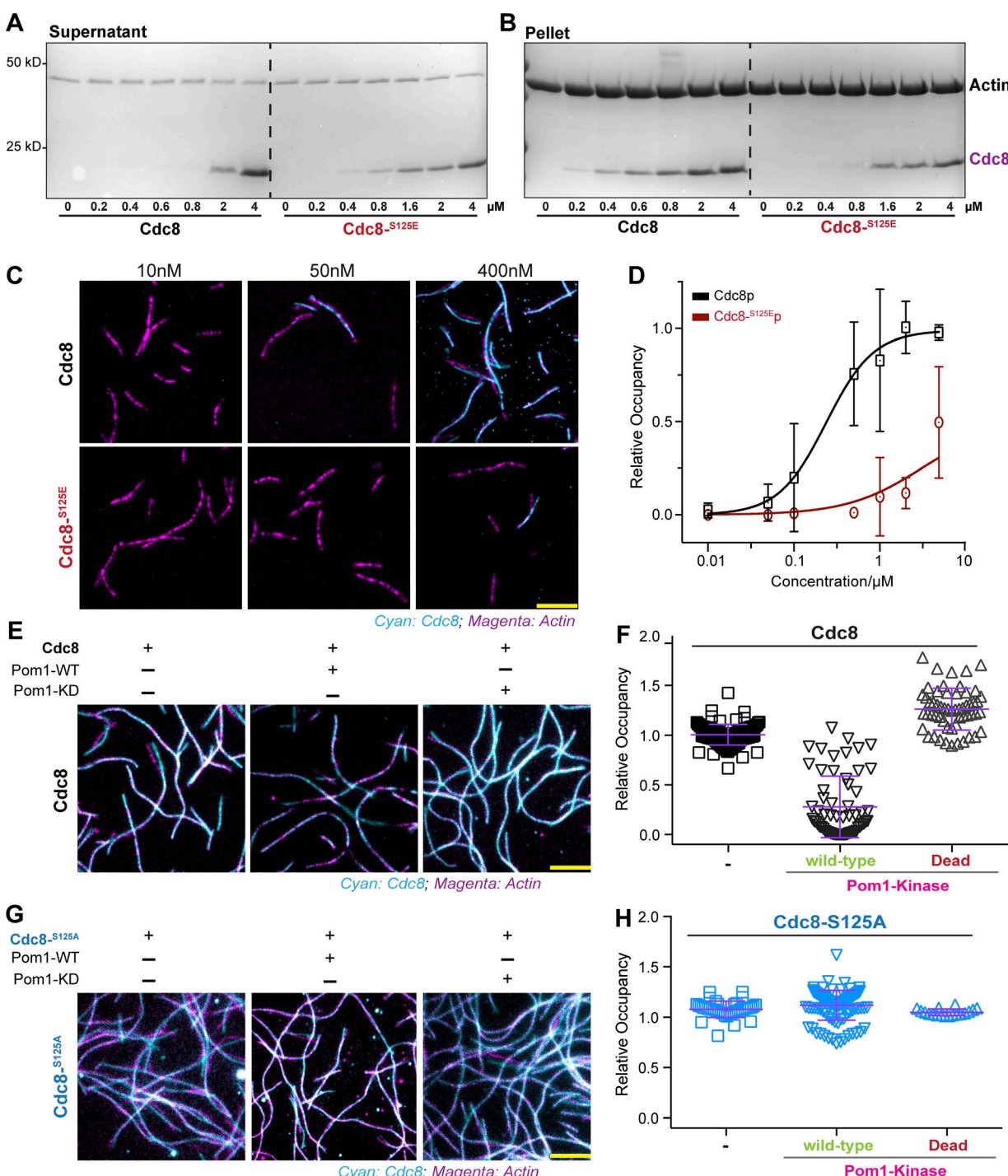

Figure 3. **Phosphorylation of S125 in Cdc8 reduces affinity to F-actin. (A)** Coomassie blue–stained gels of actin-tropomyosin cosedimentation assay using high-speed centrifugation. SDS-PAGE gel of supernatant. Each lane represents a pelleting experiment with a different concentration of ASCdc8 or ASCdc8-S125E. **(B)** Pellet fractions of ASCdc8 and ASCdc8-S125E mutants, respectively. Each lane represents a pelleting experiment with a different concentration of ASCdc8 or ASCdc8-S125E. **(C)** Image sequence showing the decoration of 125 nM F-actin-Alexa Fluor 488 (magenta) with ASCdc8-I76C-Atto-565 (top) or ASCdc8-I76C-S125E-Atto-565 (bottom) at indicated concentrations of Cdc8. Scale bar represents 5 µm. **(D)** Corresponding plot of relative F-actin decoration by ASCdc8-I76C (black) and ASCdc8-I76C-S125E (red), respectively, and fitted Hill functions (lines); WT, n = [20, 33, 46, 31, 25, 37, 6] filaments; S125E, n = [11, 9, 25, 20, 14, 27, 8] filaments. **(E)** Images showing decoration of 125 nM F-actin-Alexa Fluor 488 (magenta) with 400 nM ASCdc8-I76C-Atto-565 alone (left) or 5 min after addition of Pom1-WT (center) or Pom1-KD (right). Scale bar represents 5 µm. **(F)** Corresponding box plot of relative F-actin decoration by ASCdc8-I76C at indicated conditions; n = [119, 93, 56] fields of view. **(G)** Images showing decoration of 125 nM F-actin-Alexa Fluor 488 (magenta) with 400 nM ASCdc8-I76C-S125A-Atto-565 alone (left) or 5 min after addition of Pom1-WT (center) or Pom1-KD (right). Scale bar represents 5 µm. **(H)** Corresponding box plot of relative F-actin decoration by ASCdc8-I76C at indicated conditions; n = [40, 113, 18] fields of view. Error bars represent SD.

(Fig. 4, A and B; and Video 2). We found a similar differential effect on filament length when F-actin bound to WT ASCdc8 and ASCdc8-S125E was incubated with the actin-severing compound Swinholide-A (Bubb et al., 1995; Lim et al., 2018; Fig. 4, C and D; and Video 3). These experiments demonstrated that phosphorylation of Cdc8 causes its dissociation from actin filaments and allows access of actin-severing proteins and compounds to actin filaments.

To test the prediction that phosphorylation of Cdc8 protects actin cables from Adf1-mediated actin severing in vivo, we incubated WT, cdc8-S125A, and cdc8-S125E with the actin polymerization inhibitor Latrunculin A (LatA; Coué et al., 1987; Fig. 4 E). In WT, most actin cables and actomyosin rings were lost after 4 min of incubation with 2.5 µM LatA (Fig. 4, E–G). As expected, cells expressing Cdc8 S125E lost all actin cables after only 2 min of treatment with LatA, whereas actin cables and actomyosin rings lasted for 8 min after LatA treatment in cdc8-S125A (Fig. 4, E–G).

In this study, we provide a novel mechanism for regulation of actin filament turnover and stability that is mediated by phosphorylation of Cdc8-tropomyosin. We have shown that mutant cells expressing a phosphomimetic allele of Cdc8 have unstable actin filaments, whereas those expressing a non-phosphorylatable form of Cdc8 have more stable actin filaments. The fact that actin filaments in Cdc8-S125A undergo slower turnover suggests that *S. pombe* cells should also possess other mechanisms to facilitate actin turnover, such as myosin II–mediated actin filament breakage (Murrell and Gardel, 2012; Vogel et al., 2013), slower access of Adf1 to actin filaments, or regulation by capping proteins (Isenberg et al., 1980; Cooper and Sept, 2008; Bombardier et al., 2015). Our biochemical experiments demonstrated that phosphorylated Cdc8 has a significantly reduced affinity for F-actin, which allows the severing protein Adf1 to load on to F-actin and sever/destabilize it. We have shown that Cdc8 phosphorylation plays an important physiological role in division site placement via the tip occlusion pathway (Huang et al., 2007). Our experiments in the *mid1*-18 cells revealed that destabilization of F-actin via Cdc8 phosphorylation by the tip-localized Pom1 is an important mechanism to prevent formation of actin rings at the tip (Fig. 5). It should be noted that the tip occlusion defect in cells expressing Cdc8-S125A is weaker than in *pom1Δ* cells, suggesting that Pom1 phosphorylates also other substrates that orchestrate tip occlusion (Huang et al., 2007).

## Materials and methods

### Yeast genetics and culture methods

Mid-log-phase cells were grown and cultured in yeast extract medium (YES) at permissive temperature (24°C) as described (Moreno et al., 1991). Yeast strains carrying a single copy of Cdc8 mutants were generated by deleting the native *cdc8* gene with NATmx6 marker and integrating NotI linearized vector (plys-P*cdc8*-cdc8-Ter*cdc8*-Ura4+) carrying WT, cdc8-S125A, and cdc8-S125E at the *lys1* locus. Integrated *cdc8* mutants were under the control of native *cdc8* promoter (+500 bp before the translational start codon ATG) and *cdc8* terminator (+250 bp after the

translational termination codon TAG). Positive clones were selected on EMM-URA+NAT plates.

### PFA fixation, DAPI, anillin blue, and actin-Phalloidin staining and fluorescence microscopy

Asynchronous log-phase cells were grown at 24°C in YES before fixation. For visualization of DAPI, anillin blue/calcofluor, and Phalloidin-CF633 staining, cells were fixed with 4% PFA and permeabilized with 1% Triton X-100 at room temperature for 12 min. Cells were washed three times with 1× PBS without Triton X-100 and stained with DAPI (to visualize DNA) and anillin blue (to visualize septa and cell wall). Phalloidin-CF633 was used to visualize the actin structures (cables, patches, and rings). 27 z-stacks of 0.3-µm thickness were taken for DAPI and anillin blue. Still images were acquired at room temperature using a spinning disk confocal microscope (Andor Revolution XD imaging system, equipped with a 100× oil-immersion 1.45-NA Nikon Plan Apo lambda objective lens, confocal unit Yokogawa CSU-X1, detector Andor iXon Ultra EMCCD, and Andor iQ software). Fiji was used to process the images offline.

### Live-cell imaging

For time-lapse live-cell imaging, mid-log-phase cells were grown at 25°C. Time-lapse movies were acquired in a 25°C incubation chamber for 3–4 h. CellAsic microfluidic yeast (Y04C and Y04D) plates were used for time-lapse imaging. Time-lapse series were acquired using the spinning disk confocal microscope described above. 15 z-stacks of 0.5-µm thickness were taken for *rlc*1-3GFP and mCherry-*atb*2 at 1-min intervals. Fiji was used to process the images offline. The time of appearance of a short spindle was taken to indicate initiation of ring assembly; ring contraction was deemed to have initiated when the ring diameter started to reduce; and the time for ring contraction was the duration between initiation of ring contraction and the complete disassembly of the ring.

### Protein purification, labeling, immunoprecipitation, and Western blotting

**Cloning, mutagenesis, and protein purification.** Quick-change site-directed mutagenesis was used to create an acetylation mimicking version of Ala-Ser-Cdc8p containing phospho mutants (S125A, phosphodeficient; S125E, phosphomimetic) with and without cysteine (I76C) for protein expression. All the ASCdc8 mutants for protein expression were cloned into pETMCN vector without any tags. ASCdc8 WT and mutants were expressed in *Escherichia coli* BL21-DE3-pLysS cells for protein purification. Cultures were grown in LB media with antibiotics at 37°C and induced at an $OD_{600}$ of 0.7 by addition of 0.25 mM IPTG. Cells were induced for 4 h at 30°C and then harvested for protein purification. Pellets were resuspended in 10 ml of lysis buffer (50 mM Tris, pH 7.5, 10 mM imidazole, pH 7.5, 300 mM KCl, 5 mM EDTA, 1 mM DTT, and 5 mM MgCl₂ with 1× Roche protease cocktail inhibitors), lysed by sonication on ice, and heated to 90°C. Debris and insoluble components were removed by centrifugation at 30,000 *g* for 15 min at 4°C, and the supernatant was incubated with 10 mg/l DNase and 10 mg/l RNase at 4°C for 1 h. Cdc8 was precipitated by adjusting the pH of the supernatant to

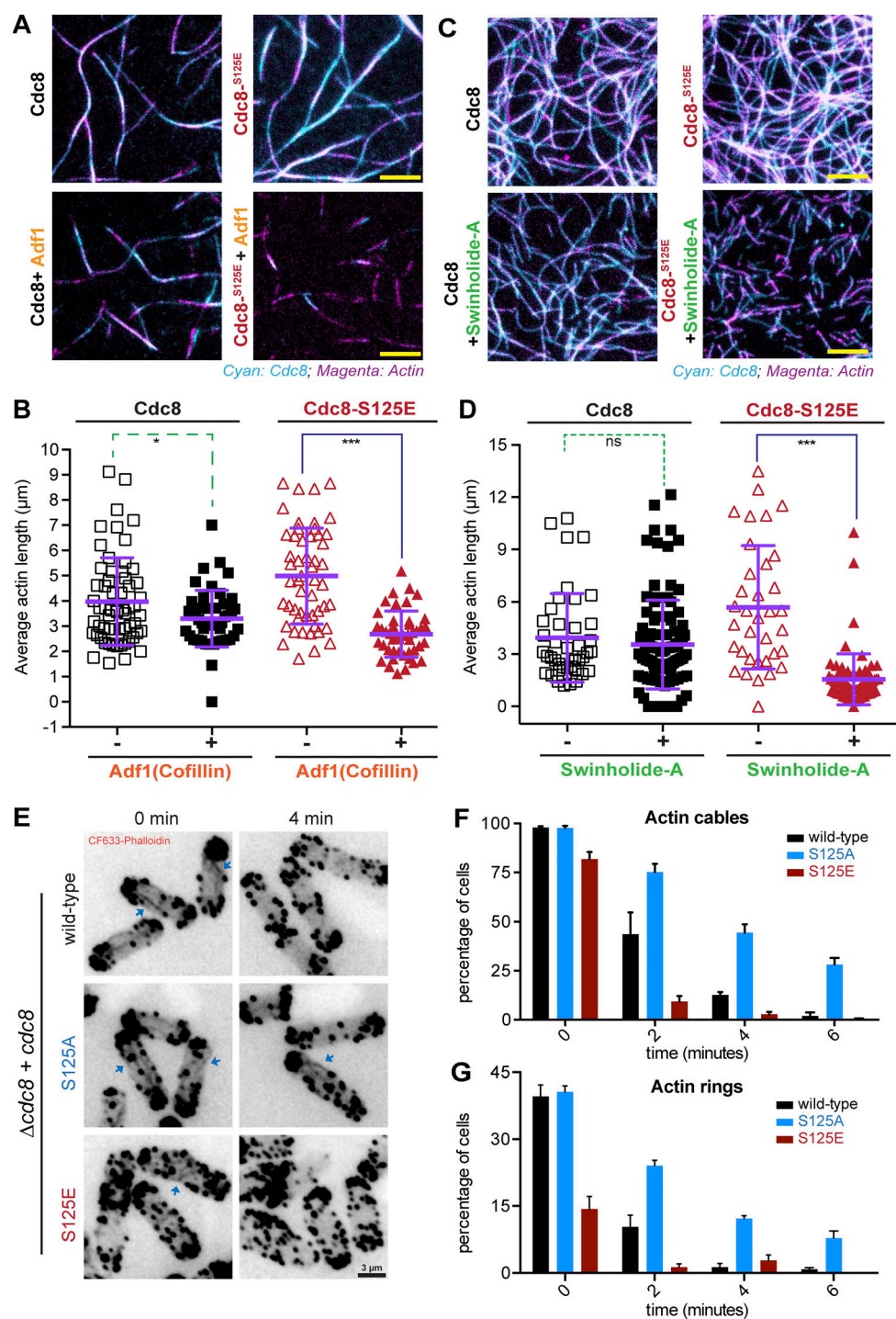

Figure 4. **Cdc8 phosphorylation on S125 facilitates Adf1-mediated F-actin disassembly in vitro and actin cable turnover in vivo. (A)** Images showing 125 nM F-actin-Alexa Fluor 488 (magenta) with 400 nM ASCdc8-I76C-Atto-565 (left) or 1,200 nM ASCdc8-I76C-S125E-Atto-565 (right) before (top) or after (bottom) addition of 100 nM Adf1/Cofilin (bottom). Scale bar represents 5 µm. **(B)** Corresponding box plot of average F-actin contour with ASCdc8-I76C (purple) and ASCdc8-I76C-S125E (red), respectively, before (hollow) and after (filled) Adf1 addition; WT, n = [74, 51] fields of view; S125E, n = [167, 221] fields of view. **(C)** Images showing 125 nM F-actin-Alexa Fluor 488 (magenta) with 400 nM ASCdc8-I76C-Atto-565 (left) or 1,200 nM ASCdc8-I76C-S125E-Atto-565 (right) before (top) or after (bottom) addition of 1,000 nM Swinholide-A (bottom). Scale bar represents 5 µm. **(D)** Corresponding box plot of average F-actin contour with ASCdc8-I76C (purple) and ASCdc8-I76C-S125E (red), respectively, before (hollow) and after (filled) Swinholide-A addition; WT, n = [41, 94] fields of view; S125E, n= [34, 76]. **(E)** cdc8+, cdc8-S125A and cdc8-S125E were treated with Lat-A (2.5 µM), and samples were taken every 2 min followed by 4% PFA fixation. Permeabilized cells were stained for actin structures with CF-633 (phalloidin). Scale bar represents 3 µm. **(F)** Quantification of E. Graph shows the fraction of cells with clearly detectable actin cables (n > 120 cells each from three independent experiments). **(G)** Quantification of E. Graph shows the fraction of cells with clearly detectable actin rings (n > 120 cells each from three independent experiments). Error bars represent SD. ns, not significant; ***, P < 0.0001; *, P < 0.017.

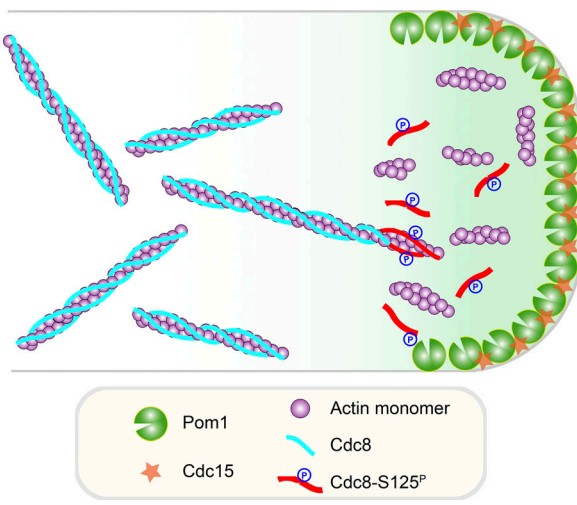

Figure 5. **A schematic representation of Pom1-mediated Cdc8 phosphorylation and its role in actin dynamics.** Pom1 kinase phosphorylates Cdc8, which causes an instability of actin filaments and inhibits actomyosin ring assembly near cell ends. See text for further details.

**Legend:**
- Pom1
- Actin monomer
- Cdc8
- Cdc15
- Cdc8-S125$^P$

the respective isoelectric focusing point of Cdc8 (pH 4.5), and the pellet was dissolved in lysis buffer with 1 mM DTT and protease inhibitor for 2–3 h at 4°C. Cdc8 was dialyzed against the storage buffer (50 mM NaCl, 10 mM imidazole, pH 7.5, and 1 mM DTT), flash frozen in liquid nitrogen, and stored at –80°C. Purified Cdc8 was reduced with 10 mM DTT at 4°C for 2 h before labeling. Cdc8 was labeled at a single cysteine residue (I76C) by atto-565 maleimide (AD 565-41, ATTO-TEC) dye per the manufacturer's protocol. Tpm$^{Cdc8}$ proteins were expressed from pJC20-based plasmids in BL21 DE3 pNatB cells to produce protein in the Nt-acetylated form (Johnson et al., 2010). Mid-log-phase cultures were grown for 3 h with 100 mg/l IPTG. Cells were harvested, resuspended in 30 ml lysis buffer (20 mM Tris, pH 7.5, 100 mM NaCl, 2 mM EGTA, and 5 mM MgCl$_2$), lysed by sonication, and heated to 85°C. Debris and insoluble components were removed by centrifugation, and the resulting supernatant was incubated with 10 mg/l DNase and 10 mg/l RNase at 4°C for 1 h. After buffer exchange into fast protein liquid chromatography (FPLC) loading buffer (5 mM Tris, pH 7.0, and 100 mM NaCl) the Tpm$^{Cdc8}$ was subjected to three rounds of FPLC purification using 2 × 5 ml Pharmacia HiTrap-Q columns in tandem, by elution with an NaCl gradient from 100 to 900 mM. After the final FPLC run, the protein was resuspended in 5 mM Tris, pH 7.0, and subjected to mass spectrometry (MS), SDS-PAGE, and spectrophotometric analyses to determine mass, purity, and protein concentration (Tpm$^{Cdc8}$ extinction coefficient, 2,980 M$^{-1}$ cm$^{-1}$).

Pom1 kinase WT (GST-Pom1-WT) and kinase-dead (GST-Pom1-KD) constructs were expressed in *E. coli* BL21-DE3-pLysS cells and purified using glutathione sepharose beads per the manufacturer's protocol (Deng et al., 2014). Adf1 was cloned into pGEX-4T1 vector, and GST-Adf1 protein was expressed in *E. coli* BL21-DE3-pLysS cells. GST-Adf1 was purified using glutathione sepharose beads per the manufacturer's protocol, and GST tag was cleaved using thrombin (GE Healthcare) before the TIRF experiment.

**Immunoprecipitation.** Mid-log-phase *S. pombe* cell pellets (WT, Δ*pom1*, and *pom1-as*, with or without inhibitor, 2 µM 3MBPP1) from a 500-ml yeast culture (10$^7$ cells/ml) were lysed in a FastPrep FP120 Cell Disturber (MP Biomedicals) using acid-washed glass beads (Sigma-Aldrich). Lysis buffer contained 50 mM Tris-HCl, pH 7.5, 300 mM KCl, 5 mM MgCl$_2$, 50 mM imidazole-HCl, pH 6.9, 0.3 mM PMSF, 350 µg/ml benzamidine, 100 mM β-glycerophosphate, 50 mM NaF, 5 mM NaVO$_3$, and complete EDTA-free protease inhibitor cocktail (Roche). Cell lysates were boiled at 90°C for 10 min and clarified at 30,000 and 50,000 $g$ for 30 min each. SpCdc8 was precipitated by 30–70% ammonium sulfate cut and dialyzed in Mono Q buffer A (50 mM KCl, 10 mM KH$_2$PO$_4$, 10 mM K$_2$HPO$_4$, 1 mM DTT, pH 7.0, and phosphatase inhibitors [100 mM β-glycerophosphate, 50 mM NaF, and 5 mM NaVO$_3$]) using a PD-midiTrap-G25 column. Purified Cdc8 (WT and S125A) was pulled down using anti-Cdc8 antibody, or anti-RxxSp was covalently coupled to Dynabeads M-270 Epoxy (Invitrogen) and magnetic beads conjugated with anti-RxxSp per the manufacturer's protocol. Purified Cdc8 proteins were incubated with the beads for 2–3 h, and beads were washed 8× with lysis buffer containing 0.5% Triton X-100. Samples were heated up for 5 min at 95°C before loading on SDS-PAGE gels. Antibodies were rabbit anti-Cdc8, anti-RxxS (Phospho-Akt Substrate [RXXS*-T*; 110B7E] rabbit mAb, Cell Signaling, NEB), goat anti-actin, and mouse anti-rabbit IgG-HRP (conformation specific [L27A9] mAb, Cell Signaling, NEB), rabbit anti-goat HRP (Cell Signaling, NEB).

**Actin purification.** Rabbit skeletal muscle actin was purified from acetone powder as described previously (Pardee and Spudich, 1982). Actin was labeled with maleimide-Alexa Fluor 488 (Molecular Probes) following the manufacturer's protocol.

**Kinase assay.** For cold kinase assay reaction, recombinant ASCdc8-WT and ASCdc8-S125A proteins were incubated with 1× Pom1 kinase buffer (30 mM Tris, 100 mM NaCl, 10 mM MgCl$_2$, 1 mM EGTA, 10% glycerol, and 40 mM ATP) with equivalent amounts of GST-Pom1-WT and GST-Pom1-KD in a 20-µl final volume reaction. After a 1-h incubation at 30°C, the reaction was stopped by boiling in sample buffer and analyzed by SDS-PAGE.

## CD

Measurements were made in 1-mm quartz cuvettes using a Jasco 715 spectropolarimeter. Tropomyosin (Tpm) proteins were diluted in CD buffer (10 mM potassium phosphate, 5 mM MgCl$_2$, pH 7.0) to a concentration of 0.4 mg/ml. CD buffer was supplemented with 500 mM NaCl (unless stated otherwise in the text) to minimize end-to-end polymerization of Tpm. Thermal unfolding data were obtained by monitoring the CD signal at 222 nm with a heating rate of 1°C · min$^{-1}$. At completion of the melting curve, the sample was cooled at a rate of 20°C · min$^{-1}$. CD spectra are presented as differential absorption (ΔA). Melting curves are reported as fraction of unfolded protein by normalizing the CD signal between 10°C and 50°C.

## Tropomyosin-actin cosedimentation assay

Cosedimentation assays were performed at 25°C by mixing 3 µM actin with increasing concentrations of ASCdc8 and ASCdc8-S125E, and then spun at 100,000 $g$ (high speed) for 20 min at

25°C. Equal volumes of supernatant and pellet were separated by 12% SDS-PAGE gel and stained with Coomassie blue (Simply-BlueStain, Invitrogen).

## Tropomyosin coating of actin filaments: TIRF microscopy

Glass coverslips (#1.5 borosilicate, Menzel) were cleaned with Hellmanex III (Hellma Analytics) according to the manufacturer's instructions, followed by thorough rinses with MilliQ water and blow dried with $N_2$. For the experimental chambers, 0.2 ml PCR tubes (Stratech) were cut to remove the lid and the conical bottom part and stuck to the cleaned glass using UV glue (NOA68, Norland Products) by 3-min curing in intense UV light at 265 nm (Stratagene UV Stratalinker 2400). This open chamber design allowed the sequential addition substances to the sample without inducing significant hydrodynamic flows that would disturb it. The surface of freshly cleaned and assembled chambers was passivated by incubation with either 1 mg/ml poly-L-lysine-polyethylene glycol 500(SuSoS; for ASCdc8 loading curves) or 0.5 mg/ml κ-casein (C0406, Sigma-Aldrich) for 20 min followed by three washes with KMEH (50 mM KCl, 2 mM $MgCl_2$, 1 mM EGTA, and 20 mM Hepes, pH 7.2), both yielding good passivation.

During this incubation time, F-actin was polymerized at a final concentration of 5 µM and a label ratio of 5%$_{mol}$. First, 10%$_{vol}$ of 10× ME buffer (100 mM $MgCl_2$ and 20 mM EGTA, pH 7.2) was mixed with unlabeled and Alexa Fluor 488–labeled G-actin in G-buffer at a concentration of 10 µM and incubated for 2 min to replace G-actin–bound $Ca^{2+}$ ions with $Mg^{2+}$. Addition of an equal volume of 2× KMEH buffer supplemented with 2 mM Mg-ATP induced F-actin polymerization at a final G-actin concentration of 5 µM. After 30-min incubation, F-actin was diluted to a concentration of 1.25 µM in KMEH supplemented with 0.5% methylcellulose and incubated for 10 min with increasing amounts of labeled ASCdc8 variants in aliquots at a final volume of 10 µl. The content of each aliquot was added to a separate experimental chamber filled with 90 µl of KMEH supplemented with 0.5% methylcellulose. Images were acquired at room temperature using a Nikon Eclipse Ti-E/B microscope equipped with a 100× oil-immersion 1.49-NA CFI Apochromat TIRF objective, a Ti-E TIRF illuminator (CW laser lines: 488, 561, and 640 nm), and a Zyla sCMOS 4.2 camera (Andor Technology) controlled by Andor iQ3 software. Fiji was used for initial image analysis and processing; see below for detailed description of quantitative image analysis.

In experiments using Pom1 kinase, F-actin incubation with ASCdc8 variants was followed by a 20-min incubation with 10 mM ATP before addition to the experimental chamber to saturate actin with ATP and prevent rapid F-actin turnover upon further addition of Pom1 and ATP. After imaging of ASCdc8-coated F-actin, 0.5 µg Pom1 and 10 mM ATP were added to the chamber and imaged.

In the experiments using F-actin–severing factor Swinholide-A (BML-T125-0020, Enzo Life Sciences), stock solutions were dissolved in DMSO (stock concentration 3 mM) and prediluted and added to chambers containing ASCdc8-coated F-actin at the final concentration (1 µM) used for the actin-severing TIRF experiments.

## Quantification and analysis

### Actin cable and patch segmentation and quantification

To quantify the amount of actin in cables and patches in *S. pombe*, we computed maximum-intensity projections from z-stacks of phalloidin-CF633–stained cells and segmented actin cables and patches using an Ilastik routine (https://ilastik.org/; Sommer et al., 2011) that was trained by manual selections in two example images. The cable-to-patch ratio was then computed by dividing the area of cables by the area of patches for each image.

### Quantification of F-actin decoration by Cdc8 and F-actin length

For generation of the actin loading graphs (Fig. 3 D), the ASCdc8 decoration length and F-actin length were measured manually for each filament, and their ratio was plotted. For the quantification of relative occupancy (Fig. 3, F and H), the area occupied by Cdc8 and F-actin was segmented using an Ilastik routine that was trained by manual selection of exemplar Cdc8 and F-actin regions. Then, the F-actin and Cdc8 regions were skeletonized (tool in ImageJ; National Institutes of Health), and the ratio of Cdc8-associated pixels versus actin-associated pixels was computed for each image pair. Since this method computed the number of pixels with F-actin and Cdc8 signal independently, slight filament drifts during the acquisition of the two fluorescence channels did not affect the quantification. For the quantification of average F-actin lengths (Fig. 4, B and D), the area occupied by F-actin was segmented out using Ilastik, followed by skeletonization. The average area of individual segments was computed for each field of view using ImageJ.

### Quantification of septum positioning

For the detection and quantification of septum positions in cells, line scans of anillin blue–stained cells were generated with ImageJ, going from tip to tip and 50 pixels wide to ensure that the entire cell was covered. The peak positions of the resulting fluorescence intensity profiles were then detected automatically using the peak analyzer function (smoothing window size: 10 pixels, method: window search, size option: height, 20%, and width, 10% of raw data) in OriginPro 2018 (OriginLab).

### Statistical analysis

All data were assumed to be normally distributed, but this was not formally tested, as visual inspection did not suggest otherwise. All box plots represent mean ± SD unless otherwise stated. Graph generation and statistical analysis was performed with Prism 6.0 software (GraphPad) and included computation of median, SD, and statistical significance using unpaired Student's *t* test (ns, not significant; ***, P < 0.0001; **, P < 0.008; *, P < 0.017).

## MS methods and analysis

### Sample preparation

Samples from cell extracts were run in 10% SDS-PAGE minigels until the dye front was 1 cm from the bottom. The gels were washed with deionized water and stained with Coomassie blue for 15 min. The Cdc8 band (∼20–25-kD region) was cut into cubes of ∼1 mm². Gel pieces were destained twice using 50%

ethanol (Thermo Fisher Scientific) in 50 mM ammonium bicarbonate (ABC, Fluka) at 22°C for 15 min and dehydrated with 100% ethanol for an additional 5 min with shaking (650 rpm). Dehydrated gel pieces were reduced with 10 mM DTT (Sigma-Aldrich) at 56°C for 30 min, alkylated with 55 mM iodoacetamide (Sigma-Aldrich), incubated at 22°C for 20 min in the dark, washed with 50% ethanol in 50 mM ABC at 22°C for 15 min, and dehydrated with 100% ethanol for 5 min. The gel pieces were hydrated with 2.5 ng/µl of trypsin (Promega) in 50 mM ammonium bicarbonate (ABC) and incubated overnight at 37°C. Peptides were extracted from gel pieces with consecutive incubations: twice with 25% acetonitrile (ACN; Thermo Fisher Scientific) with 5-min sonication in a water bath, and then with 100% ACN with 5-min sonication. Finally, supernatants were combined in a fresh vial, dried using a vacuum centrifuge at 50°C, resuspended in 50 µl of 2% ACN and 0.1% trifluoroacetic acid (Fluka), and sonicated in a water bath for 5 min.

Samples prepared for in vitro assays were digested in solution with trypsin. An aliquot of 5 µg of protein was mixed with urea buffer (8 M urea, 50 mM Tris, and 75 mM NaCl) to a final concentration of 2 M urea. The samples were first reduced with 1 mM DTT for 60 min and then alkylated with 5.5 mM iodoacetamide for 20 min in the dark. Samples were digested with 1 µg of trypsin per 100 µg of total protein overnight at room temperature. Finally, samples were desalted using StageTip (Rappsilber et al., 2007), and peptides were prepared for MS analysis as described above for the in-gel digestion.

Liquid chromatography-MS/MS analysis was performed using an Ultimate 3000-RSLCnano system (Dionex) and an Orbitrap-Fusion (Thermo Fisher Scientific). 1 µg of peptides was loaded on an Acclaim-PepMap µprecolumn (Thermo Fisher Scientific, 300-µm internal diameter by 5-mm length, 5-µm particle size, 100-Å pore size) and equilibrated in 2% ACN and 0.1% trifluoroacetic acid for 5 min at 10 µl/min with an analytical column, Acclaim PepMap RSLC (Thermo Fisher Scientific, 75 µm internal diameter by 25-cm length, 2-µm particle size, 100-Å pore size). Mobile phase A was 0.1% formic acid, and mobile phase B was ACN containing 0.1% formic acid. Peptides were eluted at 300 nl/min by increasing mobile phase B from 8% to 25% over 14 min, to 35% B over 2 min, to 90% B for 1 min, and 8-min reequilibration at 3% B. MS data were acquired with Xcalibur v3.0.63 (Thermo Fisher Scientific). Electrospray used a static Nanospray-Flex with a stainless-steel emitter, OD 1/32, in positive mode at 2.1 kV (Thermo Fisher Scientific). MS survey scans from 375 to 1,500 $m/z$, with a $4 \times 10^5$ ion count target, maximum injection time of 200 ms, and resolution of 240,000 at 200 $m/z$ were acquired in profile mode performed in the Orbitrap analyzer. Data-dependent mode selected the most abundant precursor ions possible in a 2-s cycle time followed by 45-s exclusion, and ions were isolated in the quadrupole with a 1.2-$m/z$ window. MS/MS scans were performed in the ion trap in rapid mode with ion count target of $10^4$, maximum injection time of 200 ms, and acquired in centroid mode. Precursor ions were fragmented with higher energy C-trap dissociation, normalized collision energy of 30%, and fixed first mass of 120 $m/z$. Thermo Fisher Scientific raw files were analyzed using MaxQuant software v1.6.0.16 (Tyanova et al., 2016a,b) against the

UniProtKB *S. pombe* database (UP000002485, 5,142 entries, release January 2017). Peptide sequences were assigned to MS/MS spectra using the following parameters: cysteine carbamidomethylation as a fixed modification; protein N-terminal acetylation, methionine oxidations, and S, T, Y phosphorylation as variable modifications. The false discovery rate was set to 0.01 for both proteins and peptides, with a minimum length of seven amino acids and was determined by searching a reversed database. Enzyme specificity was trypsin with a maximum of two missed cleavages. Peptide identification was performed with an initial precursor mass deviation of 7 ppm and a fragment mass deviation of 20 ppm. The MaxQuant feature "match between runs" was enabled, and label-free protein quantification was done with a minimum ratio count of 2. Scaffold (v4.6.2, Proteome Software) was used to validate MS/MS-based peptide and protein identifications. Peptide identifications were accepted if they could be established at >90.0% probability by the Scaffold local false discovery rate algorithm. Protein identifications were accepted if they could be established at >90.0% probability and contained at least two identified peptides. Proteins that contained similar peptides and could not be differentiated based on MS/MS analysis alone were grouped to satisfy the principles of parsimony. Proteins sharing significant peptide evidence were grouped into clusters.

### Online supplemental material

Fig. S1 shows actin cables structures in cdc8, cdc8-S125A, and cdc8-S125E, stained with antibodies against Cdc8. Fig. S2 shows that mid1-18 cdc8-S125A mutant cells form tip septa at restrictive temperature. Fig. S3 shows that Cdc8 S125A and Cdc8 S125E mutants are functional at higher temperatures, generation of functional fluorescent reporters for TIRF experiments, and that Cdc8 is a substrate of Pom1 kinase. Table S1 lists all yeast strains used in this study. The videos show TIRF imaging of Cdc8 and Cdc8-S125A coated F-actin after addition of GST-Pom1-WT and GST-Pom1-KD (Video 1), Cdc8 and Cdc8-S125E coated F-actin after addition of Adf1 (Cofilin; Video 2), and actin alone or Cdc8/Cdc8-S125E coated F-actin after addition of the actin-severing drug (Swinholide-A; Video 3).

### Acknowledgments

We thank James Moseley (Dartmouth University, Hanover, NH), Sarah E. Hitchcock-DeGregori (Rutgers University, Piscataway, NJ), and Sophie Martin (University of Lausanne, Lausanne, Switzerland) for reagents and strains used in this work.

M.K. Balasubramanian was supported by Wellcome Trust Senior Investigatorship (WT101885MA), Wellcome Trust Collaborative Award (203276/Z/16/Z), Royal Society Wolfson Merit Award (WM130042), and European Research Council Advanced Grant (ERC-2014-ADG No. 671083). J.B.A. Millar was supported by Medical Research Council program grant MR/K001000/1.

The authors declare no competing financial interests.

Author contributions: S. Palani and M.K. Balasubramanian conceived and designed the project. S. Palani designed and performed most of the experiments. S. Palani and D.V. Köster designed and performed TIRF experiments and edited the

manuscript. S. Palani and T. Hatano performed experiments for Fig. 2 (E–J) and Fig. 3 (A and B). A. Kamnev performed initial TIRF experiments for Fig. 3 (C and D). H.R. Brooker and D.P. Mulvihill performed experiments for Fig. S3 (A–D) and edited the manuscript. S. Palani, J.R. Hernandez-Fernaud, and A.M.E. Jones performed and analyzed MS phosphorylation data for Fig. 1 B and edited the manuscript. T. Kanamaru performed initial experiments and generated reagents. J.B.A. Millar identified Cdc8 phosphorylation in unpublished work. S. Palani and M.K. Balasubramanian wrote the manuscript with input from other authors, notably J.B.A. Millar and D.V. Köster.

Submitted: 17 September 2018

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
