## [Reviewer comments · The Journal of Cell Biology]

Phospho-regulation of tropomyosin is crucial for actin cable turnover and division site placement

Saravanan Palani, Darius Köster, Tomoyuki Hatano, Anton Kamnev, Taishi Kanamaru, Holly Brooker, Juan Ramon Hernandez-Fernaud, Alexandra Jones, Jonathan Millar, Daniel Mulvihill, and Mohan Balasubramanian

Corresponding Author(s): Mohan Balasubramanian, University of Warwick and Saravanan Palani, Warwick Medical School

Review Timeline:

Submission Date:	2018-09-17
Editorial Decision:	2018-09-21
Revision Received:	2018-11-29
Editorial Decision:	2019-01-07
Revision Received:	2019-07-23
Editorial Decision:	2019-08-19
Revision Received:	2019-08-27

Monitoring Editor: Daniel Lew

Scientific Editor: Tim Spencer

Transaction Report:

DOI: <https://doi.org/N/A>

September 21, 2018

Re: JCB manuscript #201809089

Prof. Mohan K Balasubramanian
University of Warwick
Warwick Medical School The University of Warwick
Coventry CV4 7AL
United Kingdom

Dear Prof. Balasubramanian,

Thank you for submitting your Report manuscript entitled "Phospho-regulation of tropomyosin is crucial for actin cable turnover in fission yeast" to Journal of Cell Biology. As part of our normal reviewing procedure, your paper has been evaluated by at least two editors and an editorial statement is provided below. You will see that, in the consensus opinion of our editors, although we are interested in the concepts presented in this study, the manuscript is too preliminary for external review. We have thus decided not to subject your manuscript to a lengthy review process. We would be willing to consider a revised manuscript containing data addressing the detailed editorial comments below, assuming the novelty of the findings has not been compromised in the interim.

Because Journal of Cell Biology addresses a wide and diverse audience of cell biologists, we must give priority to manuscripts that provide a substantial advance of broad appeal to the cell biology community, even though many others also present interesting and important advances for researchers in a particular field.

I am sorry that our answer on this occasion is not more positive, and I hope that this outcome will not dissuade you from submitting other manuscripts to us in the future.

Thank you for your interest in Journal of Cell Biology.

With kind regards,

Jodi Nunnari
Editor-in-Chief
Journal of Cell Biology

Editorial Statement:

In this study, Palani et al report that in contrast to muscle cells where the interaction between tropomyosin and F-actin is regulated by calcium, fission yeast tropomyosin is regulated via phosphorylation at S125. They show that phosphorylated tropomyosin has reduced affinity for F-actin, allowing the severing protein Adf1 to destabilize filaments. The study appears carefully done, having examined both phosphomimetic mutants as well as the effect of phosphorylation. However, to be considered as a Report in JCB a study must represent a highly novel finding of interest to a broad cell biology audience. Showing that tropomyosin can be regulated by phosphorylation is

certainly of interest to researchers in the specific field, but some indication of the physiological significance of the phosphorylation would make it of broader interest to the JCB readership. Therefore, we would welcome and send out for review a study that assessed the physiological role(s) of S125 phosphorylation (e.g. using the non-phosphorylatable mutant). For example, might roles in interphase cable function/reorganization be revealed in suitable genetic backgrounds? Might the phosphorylation-antibody be used to ask when in the cell cycle Cdc8 phosphorylation occurs? This may suggest roles in NETO or polarized growth rather than cytokinesis. If Pom1 does phosphorylate Cdc8 in vivo, might one expect that to occur near cell poles where Pom1 is enriched?

January 7, 2019

Re: JCB manuscript #201809089R-A

Prof. Mohan K Balasubramanian
University of Warwick
Warwick Medical School The University of Warwick
Coventry CV4 7AL
United Kingdom

Dear Prof. Balasubramanian,

Thank you for submitting your manuscript entitled "Phospho-regulation of tropomyosin is crucial for actin cable turnover and division site placement". The manuscript was assessed by expert reviewers, whose comments are appended to this letter. We invite you to submit a revision if you can address the reviewers' key concerns, as outlined here.

You will see that while the reviewers are overall positive about your findings, they have made constructive suggestions to further improve your study. In particular, a main experimental issue in revision should be to strengthen the evidence for Pom1-mediated phosphorylation of S125. In addition, we hope that you will be able to address all of the remaining reviewer comments in your revised manuscript.

GENERAL GUIDELINES:

Text limits: Character count for a Report is < 20,000, not including spaces. Count includes title page, abstract, introduction, results, discussion, acknowledgments, and figure legends. Count does not include materials and methods, references, tables, or supplemental legends.

Figures: Reports may have up to 5 main text figures. To avoid delays in production, figures must be prepared according to the policies outlined in our Instructions to Authors, under Data Presentation, <http://jcb.rupress.org/site/misc/ifora.xhtml>. All figures in accepted manuscripts will be screened prior to publication.

Supplemental information: There are strict limits on the allowable amount of supplemental data. Reports may have up to 3 supplemental figures. Up to 10 supplemental videos or flash animations are allowed. A summary of all supplemental material should appear at the end of the Materials and methods section.

Our typical timeframe for revisions is three months; if submitted within this timeframe, novelty will not be reassessed at the final decision. Please note that papers are generally considered through only one revision cycle, so any revised manuscript will likely be either accepted or rejected.

Thank you for this interesting contribution to Journal of Cell Biology. You can contact us at the journal office with any questions, cellbio@rockefeller.edu or call (212) 327-8588.

Sincerely,

Daniel Lew
Monitoring Editor

Andrea L. Marat
Scientific Editor

Journal of Cell Biology

Reviewer #1 (Comments to the Authors (Required)):

Tropomyosins stabilize actin filaments in most eukaryotic cells. This paper studies regulation of tropomyosin in fission yeast cells, with the goal of understanding mechanisms that regulate polarized actin cables and the cytokinetic actin ring. Past studies have shown that tropomyosin and Adf1/cofilin compete for binding to actin filaments, so mechanisms that regulate either protein could shift their competition at specific sites and times in cells. Here, the authors characterize phosphorylation at Ser125 on fission yeast tropomyosin (cdc8). Through non-phosphorylatable and phosphomimetic mutants, they show that pSer125 reduces actin cable abundance during interphase and prevents actin ring assembly at cell tips during division. Biochemical experiments show that the cell polarity kinase Pom1 may phosphorylate Cdc8-Ser125 to limit interaction with actin filaments, thus allowing Adf1/cofilin to sever filaments. In this manner, phosphorylation at Ser125 represents a novel mechanism to regulate the balance of actin stabilization by Cdc8 versus actin destabilization by Adf1. Overall, this paper uses a range of techniques to identify and characterize a novel mechanism. It is a very strong and interesting paper, although I do see a number of places where improvements could be made. I have listed specific comments below, and I feel that the paper will be substantially improved by addressing these comments. The large number of comments does not reflect a lack of enthusiasm for the paper.

1. I am confused about the source of the data in Figure 1B. The authors describe mass spec experiments in the methods section but not in the main text. Some additional description of the experiment that generated this data panel would be helpful.
2. The authors state that there are no cell cycle fluctuations in Cdc8-pSer125, but no data are shown for this statement. They should provide the data or remove the statement.
3. The actin cables in Figure 1F (and other figures) are not very obvious. The Balasubramanian lab

- has established Lifeact as an excellent marker for actin structures in fission yeast, and this marker seems to highlight cables better than phalloidin, at least comparing the images in this paper to the images from their lab's past publications. Since the cable phenotype is not very clear to my eye, it would be helpful to try repeating this experiment with Lifeact.
4. The authors refer to a 'split septa' phenotype in Figure 2A, but need to measure the frequency of this phenotype in the population.
 5. In the text describing results for Figures 2C and D, the authors provide qualitative descriptions that could be strengthened by quantifying their phenotypes. For example, the "delay in ring assembly (Fig. 2C) should be measured for 10+ cells. They also provide rough numbers for 'significantly longer' actomyosin ring contraction in the *cdc8-125E* mutant versus wild type. This timing should be measured for multiple cells and reported as mean plus/minus standard deviation.
 6. What is the phenotype of *mid1-18 cdc8-S125E* cells? Figure 2 provides a nice description of *mid1* mutant combined with *cdc8*-alanine mutant, but the glutamate mutant is missing from this analysis.
 7. The authors purify acetylated Cdc8 protein, but then switch to using an acetyl mimic for their in vitro experiments. Why not use the true acetylated protein for these experiments?
 8. Some additional experiments would provide great insight into how phosphorylation inhibits Cdc8 interaction with actin. For example, in Figure 3, the Cdc8-125E mutant pelleting with actin is not extended to saturating concentrations. It seems important to know if the mutant saturates at the same Cdc8:actin stoichiometry as the wild type. Further, by altering the timing of Pom1 addition, the authors could determine if phosphorylation prevents association of Cdc8 with actin, or alternatively if phosphorylation promotes dissociation.
 9. The authors propose that Pom1 phosphorylates Cdc8-Ser125 based on functional studies, but more direct experiments are needed. They should perform two experiments that use reagents already in hand. First, they need to perform in vitro kinase assays to demonstrate direct phosphorylation of purified Cdc8 by Pom1. Second, they need to test how pom1 mutations affect the level of Cdc8-S125 phosphorylation in cells, using an experiment such as in Figure 1D.
 10. Were experiments in Figure 4F-G performed multiple times? This result could be strengthened with replicates.
 11. The localization of Cdc8-S125A and Cdc8-S125E in cells would add a lot of insight to the mechanism. For example, is Cdc8-S125A at actin patches, and is Cdc8-S125E absent from cables and reduced at rings?
 12. Some additional discussion of the mechanism in cells would add perspective to the paper. For example, why does phosphorylation by Pom1 at cell ends inhibit actin cable assembly? This seems counterintuitive given the role of Pom1 in cell polarity, and the assembly of cables at cell ends. One possibility would be that Adf1 needs to 'trim' actin filaments that are subsequently assembled into multi-filament cables. Other possibilities could also be envisioned, and it would be helpful to the reader if the authors can provide such bigger picture models. Similarly, do the authors envision a phosphatase acting in the cytoplasm or also at cell tips? Defects in the S125E mutant indicate that something limits phosphorylation in cells to prevent its accumulation to toxic levels.

Reviewer #2 (Comments to the Authors (Required)):

Review of Palani et al for JCB

In this manuscript, Palani et al describe phospho-regulation of tropomyosin in fission yeast. They show that tropomyosin is phosphorylated on serine 125, which leads to reduced affinity with F-actin and increased access of severing protein cofilin. They identify the likely kinase as Pom1, a kinase that contributes to medial division site placement. They propose this forms part of the mechanism

by which fission yeast cells prevents assembly of division ring at cell poles.

I enjoyed reading this manuscript, which is clearly written and largely convincing. The only claim that could be better supported is the link between Pom1 kinase and Cdc8 phosphorylation. I also have a few additional comments below that the authors may want to address to strengthen the manuscript before publication.

In Figure 3G, the decoration of actin filaments by Cdc8-S125A appears less convincing in presence of Pom1-WT than Pom1-KD. This may be because of slight movement of actin filaments between image acquisition, but is somewhat unfortunate as it is a key experiment to establish that Pom1 is a critical kinase for S125. Could you find a more convincing example? The quantification method is also not very clear, as the method section indicates two alternative quantification strategies (and refers only to panel 4B, not 3G). Could you specify which one is used and how slight movement between fluorescence channel acquisition are taken into account?

The evidence that Pom1 is a bona fide kinase for Cdc8 S125 could be further supported. Is Pom1 phosphorylating Cdc8 directly in vitro on S125? Is Cdc8 phosphorylation in vivo Pom1-dependent? Does the Cdc8-S125E mutant suppress the tip occlusion defect of pom1 mutants?

The number of tip-positioned rings is somewhat different using phalloidin staining or Rlc1-GFP. Does Rlc1-GFP have a deleterious effect? It also looks like the % of septa at cell ends is much lower than the % of rings at cell ends. Are rings sliding back towards cell middle, or not leading to septa formation?

The role and use of acetylated vs. acetylation-mimicking Cdc8 alleles relative to the phosphorylation status is unclear. In the in vitro assay, it is first explained that acetylated Cdc8 is purified (though I cannot find methods explaining this purification), but then that acetylation mimicking version of Cdc8 is used. Can you clarify?

Minor comments:

S125 phosphorylation should be stated in the first sentence of the manuscript. As is, when first introduced in the text, it is currently not said to be phosphorylated.

The growth of cdc8-5E mutant in Fig S1 looks somewhat different from that of the S125E single mutant shown in Fig 1, contrary to what is claimed in the text. I understand one is done in a cdc8 Δ background, the other in a cdc8-ts mutant, which may be the cause of the difference. Showing growth of comparable strains on the same plate of single and 5-fold mutant would be better, and/or change the text.

In quantifying cytokinesis defects, what reference point is used to measure the time of ring assembly and contraction?

In the text describing Fig 4A, "...displayed a significant reduction in F-actin length", it would be helpful to write upon addition of Adf1. Also, in the quantification of this experiment, μm is missing on the label to the graph y-axis and statistics is missing.

Reviewer #1 (Comments to the Authors (Required)):

Tropomyosins stabilize actin filaments in most eukaryotic cells. This paper studies regulation of tropomyosin in fission yeast cells, with the goal of understanding mechanisms that regulate polarized actin cables and the cytokinetic actin ring. Past studies have shown that tropomyosin and Adf1/cofilin compete for binding to actin filaments, so mechanisms that regulate either protein could shift their competition at specific sites and times in cells. Here, the authors characterize phosphorylation at Ser125 on fission yeast tropomyosin (*cdc8*). Through non-phosphorylatable and phosphomimetic mutants, they show that pSer125 reduces actin cable abundance during interphase and prevents actin ring assembly at cell tips during division. Biochemical experiments show that the cell polarity kinase Pom1 may phosphorylate Cdc8-Ser125 to limit interaction with actin filaments, thus allowing Adf1/cofilin to sever filaments. In this manner, phosphorylation at Ser125 represents a novel mechanism to regulate the balance of actin stabilization by Cdc8 versus actin destabilization by Adf1. Overall, this paper uses a range of techniques to identify and characterize a novel mechanism. It is a very strong and interesting paper, although I do see a number of places where improvements could be made. I have listed specific comments below, and I feel that the paper will be substantially improved by addressing these comments. The large number of comments does not reflect a lack of enthusiasm for the paper.

We thank the referee for his/her constructive and enthusiastic comments. We have fully addressed the concerns and hope the paper will be acceptable for publication in JCB. Our responses are in blue font.

1. I am confused about the source of the data in Figure 1B. The authors describe mass spec experiments in the methods section but not in the main text. Some additional description of the experiment that generated this data panel would be helpful.

RESPONSE: The data in Figure 1B comes from our own proteomic experiments, which confirm the data from the published whole proteome studies. We have made this clear in the manuscript. In short, Cdc8 was purified as a heat stable protein at 95°C in 1M KCl and subject to proteomic analysis. This has also been clearly mentioned in the manuscript.

2. The authors state that there are no cell cycle fluctuations in Cdc8-pSer125, but no data are shown for this statement. They should provide the data or remove the statement.

RESPONSE: We have removed this sentence from the manuscript.

3. The actin cables in Figure 1F (and other figures) are not very obvious. The Balasubramanian lab has established Lifeact as an excellent marker for actin structures in fission yeast, and this marker seems to highlight cables better than phalloidin, at least comparing the images in this paper to the images from their lab's past publications. Since the cable phenotype is not very clear to my eye, it would be helpful to try repeating this experiment with Lifeact.

RESPONSE: We have provided better images, which adequately convey the message in the revised manuscript. In addition, in Figure S1, we provide images of wild-type, *cdc8-S125A*, and *cdc8-S125E* cells stained with Cdc8Abs, which again clearly demonstrates the presence of increased numbers of actin cables in *cdc8-S125A*. We have not used the Lifeact tool, since it appeared to rescue the *cdc8 S125E* phenotype partially, making it difficult to evaluate the defects in *cdc8 S125E* mutants.

4. The authors refer to a 'split septa' phenotype in Figure 2A, but need to measure the frequency of this phenotype in the population.

RESPONSE: The quantitation has been included in the legend to figure 2A.

5. In the text describing results for Figures 2C and D, the authors provide qualitative descriptions that could be strengthened by quantifying their phenotypes. For example, the "delay in ring

assembly (Fig. 2C) should be measured for 10+ cells. They also provide rough numbers for 'significantly longer' actomyosin ring contraction in the *cdc8-125E* mutant versus wild type. This timing should be measured for multiple cells and reported as mean plus/minus standard deviation.

RESPONSE: These statistical data had been provided in the original manuscript, but have now been made clear in the revised manuscript ("*cdc8-S125E* had abnormal actomyosin rings whose assembly took significantly longer than in wild-type (Fig. 2C and D; 19.6 ± 1.3 minutes in wild-type compared to 42.7 ± 5.8 minutes in *cdc8-S125E*). In addition, actomyosin ring contraction was aberrant and took significantly longer in *cdc8-S125E* (53 ± 17 minutes) compared to wild type (31 ± 2.7 minutes) (Fig. 2D)").

6. What is the phenotype of *mid1-18 cdc8-S125E* cells? Figure 2 provides a nice description of *mid1* mutant combined with *cdc8*-alanine mutant, but the glutamate mutant is missing from this analysis.

RESPONSE: This is a very interesting point. We have spent a great deal of time and effort in trying to make the mutant *mid1-18 cdc8::NatMX6 cdc8S125E*, but we have not succeeded in obtaining this mutant due to synthetic lethality of this combination. However, we succeeded in making a *mid1-18 cdc8-110 cdc8 S125E* mutant and found that the *cdc8 S125E* mutant in this *mid1*-defective background resembles the *cdc8-110* at the restrictive temperature. We have included some images of these cells for the benefit of the referee, but have not included them in the manuscript, since the comparisons will not be appropriate, with *cdc8 S125A* being in the *cdc8*-null background and the *cdc8 S125E* being in the *cdc8-110* background.

Reviewer:1 (Q6)

7. The authors purify acetylated Cdc8 protein, but then switch to using an acetyl mimic for their in vitro experiments. Why not use the true acetylated protein for these experiments?

RESPONSE: We have used the acetyl mimic version for the single molecule TIRF microscopy based studies so as to be consistent with work being done by other researchers in the field (such as David Kovar and Kathy Trybus), who have used acetyl mimic versions. We believe this way the field will have a common set of parameters in studies of the actin cytoskeleton *in vitro*. In addition, we also believe that the use of the acetyl mimic version ensures that the entire population of the Cdc8 is homogenous, unlike when acetylation is used, in which it may be possible that only a fraction is acetylated.

8. Some additional experiments would provide great insight into how phosphorylation inhibits Cdc8 interaction with actin. For example, in Figure 3, the Cdc8-125E mutant pelleting with actin is not extended to saturating concentrations. It seems important to know if the mutant saturates at the same Cdc8:actin stoichiometry as the wild type. Further, by altering the timing of Pom1 addition, the authors could determine if phosphorylation prevents association of Cdc8 with actin, or alternatively if phosphorylation promotes dissociation.

RESPONSE: We have carried out this experiment and show in the revised manuscript that Cdc8 and Cdc8S125E bind actin equally efficiently at saturating concentration of $8 \mu\text{M}$.

The Reviewer raises a very interesting point here, whether phosphorylation affects K_{on} , K_{off} or both. Our current data with Cdc8-S125E already indicates that phosphorylation alters the association rate of Cdc8 to actin, as the cooperative build up of Cdc8 decoration only occurs at concentrations 4 times higher than with wild-type Cdc8. Preincubating Cdc8 with Pom1 prior to their addition to F-actin would be equivalent to our experiments with Cdc8-S-125E. We did not attempt this experiment, as experimental limitations (Pom1 in excess, high ATP concentrations to ensure Pom1 activity) would make it very unlikely to distinguish whether Pom1 phosphorylates Cdc8 in solution and hence, prevents it from binding to actin or whether Pom1 acts on Cdc8 at the moment it binds to actin and causes its dissociation, as we wouldn't be able to know, whether the Cdc8 binding to actin would be phosphorylated or not. The best experiment to conduct here, would be to test for the Pom1 activity on Cdc8 in presence and absence of actin, but this would go beyond of the scope of this study and might be part of future work.

9. The authors propose that Pom1 phosphorylates Cdc8-Ser125 based on functional studies, but more direct experiments are needed. They should perform two experiments that use reagents already in hand. First, they need to perform *in vitro* kinase assays to demonstrate direct phosphorylation of purified Cdc8 by Pom1. Second, they need to test how pom1 mutations affect the level of Cdc8-S125 phosphorylation in cells, using an experiment such as in Figure 1D.

RESPONSE: We have provided additional evidence for this conclusion in two ways, as suggested by the referee. 1. We have shown that Phospho-Cdc8 is reduced in *pom1Δ* and *pom1as* mutants and 2. We have shown that bacterially expressed Pom1 phosphorylates Cdc8 (but not Cdc8 S125A) *in vitro*.

10. Were experiments in Figure 4F-G performed multiple times? This result could be strengthened with replicates.

RESPONSE: These experiments have been performed 3 times and the information is provided in the legend to figure 4F and G.

11. The localization of Cdc8-S125A and Cdc8-S125E in cells would add a lot of insight to the mechanism. For example, is Cdc8-S125A at actin patches, and is Cdc8-S125E absent from cables and reduced at rings?

RESPONSE: We have performed this experiment and we did not see Cdc8S125A in actin patches in immunofluorescence experiments. The ring localized Cdc8 was very faint in the Cdc8 S125E cells, but whether the fluorescence intensity is a cause or consequence is unclear. Thus, we have not included this information in the manuscript. The images are provided for the referees' benefit.

12. Some additional discussion of the mechanism in cells would add perspective to the paper. For example, why does phosphorylation by Pom1 at cell ends inhibit actin cable assembly? This seems counterintuitive given the role of Pom1 in cell polarity, and the assembly of cables at cell ends. One possibility would be that Adf1 needs to 'trim' actin filaments that are subsequently assembled into multi-filament cables. Other possibilities could also be envisioned, and it would be helpful to the reader if the authors can provide such bigger picture models. Similarly, do the authors envision a phosphatase acting in the cytoplasm or also at cell tips? Defects in the S125E mutant indicate that something limits phosphorylation in cells to prevent its accumulation to toxic levels.

RESPONSE: In the revised discussion, we have marginally expanded the discussion about the possible mechanism of Pom1 at the cell ends. In short, we believe that Pom1 action keeps actin filaments free from Cdc8 and hence enables their trimming by adf1, which in turn provides fresh g-actin for incorporation into actin patches at the cell poles. The issue pertaining to the phosphatase is interesting, but in the interest of space we are unable to discuss it at any length.

Reviewer #2 (Comments to the Authors (Required)):

Review of Palani et al for JCB

In this manuscript, Palani et al describe phospho-regulation of tropomyosin in fission yeast. They show that tropomyosin is phosphorylated on serine 125, which leads to reduced affinity with F-actin and increased access of severing protein cofilin. They identify the likely kinase as Pom1, a kinase that contributes to medial division site placement. They propose this forms part of the mechanism by which fission yeast cells prevents assembly of division ring at cell poles.

I enjoyed reading this manuscript, which is clearly written and largely convincing. The only claim that could be better supported is the link between Pom1 kinase and Cdc8 phosphorylation. I also have a few additional comments below that the authors may want to address to strengthen the manuscript before publication.

1. In Figure 3G, the decoration of actin filaments by Cdc8-S125A appears less convincing in presence of Pom1-WT than Pom1-KD. This may be because of slight movement of actin filaments between image acquisition, but is somewhat unfortunate as it is a key experiment to establish that Pom1 is a critical kinase for S125. Could you find a more convincing example? The quantification method is also not very clear, as the method section indicates two alternative quantification strategies (and refers only to panel 4B, not 3G). Could you specify which one is used and how slight movement between fluorescence channel acquisition are taken into account?

RESPONSE: We have provided better images for the decoration of actin filaments with Cdc8 S125A. The quantification details have been provided as requested.

2. The evidence that Pom1 is a bona fide kinase for Cdc8 S125 could be further supported. Is Pom1 phosphorylating Cdc8 directly in vitro on S125? Is Cdc8 phosphorylation in vivo Pom1-dependent? Does the Cdc8-S125E mutant suppress the tip occlusion defect of pom1 mutants?

RESPONSE: The referee has asked for three lines of evidence and we have provided two of these. In the revised manuscript, we have shown that bacterially expressed and purified Pom1 phosphorylates Cdc8, but not Cdc8 S125A. We have also shown that Cdc8 phosphorylation is reduced in *pom1Δ* and *pom1as* mutants. For technical reasons, we have been unable to get *pom1Δ cdc8Δ cdc8-S125E mid1-18*, in which tip occlusion defects can be investigated (please see response to point 6 of referee 1 as well).

3. The number of tip-positioned rings is somewhat different using phalloidin staining or Rlc1-GFP. Does Rlc1-GFP have a deleterious effect? It also looks like the % of septa at cell ends is much lower than the % of rings at cell ends. Are rings sliding back towards cell middle, or not leading to septa formation?

RESPONSE: The number of rings using phalloidin and Rlc1 are similar and the seeming differences in our original submission were due to different scaling of the y-axis in the Rlc1 and phalloidin experiments. Our apologies for this confusion. These have now been corrected.

The reduced frequency of septa compared to rings at cell ends likely results from additional potential mechanisms that prevent tip-septation, such as Pom1 regulation via Cdc15 and other possible substrates. These have been discussed in the manuscript.

4. The role and use of acetylated vs. acetylation-mimicking Cdc8 alleles relative to the phosphorylation status is unclear. In the in vitro assay, it is first explained that acetylated Cdc8 is purified (though I cannot find methods explaining this purification), but then that acetylation mimicking version of Cdc8 is used. Can you clarify?

RESPONSE: Although we used acetylated Cdc8 in structural studies involving circular dichroism and melting temperature analyses, we have used acetyl mimic mutants in all other experiments.

The reason was to generate data with tools similar to those used by the laboratories of David Kovar and Kathy Trybus as well as the ease of expression of the Acetyl mimic form. Please see response to point 7 of referee 1.

Minor comments:

1. S125 phosphorylation should be stated in the first sentence of the manuscript. As is, when first introduced in the text, it is currently not said to be phosphorylated.

RESPONSE: We do not understand this point, but have clearly started the first sentence of the results with a mention of phosphorylation of Cdc8.

2. The growth of *cdc8-5E* mutant in Fig S1 looks somewhat different from that of the S125E single mutant shown in Fig 1, contrary to what is claimed in the text. I understand one is done in a *cdc8Δ* background, the other in a *cdc8-ts* mutant, which may be the cause of the difference. Showing growth of comparable strains on the same plate of single and 5-fold mutant would be better, and/or change the text.

RESPONSE: The referee is right in that the strains were in different backgrounds. Since the data does not add much to the story and in the interest of space, we have removed description of the quintuple mutant from the revised manuscript.

3. In quantifying cytokinesis defects, what reference point is used to measure the time of ring assembly and contraction?

RESPONSE: The timing of appearance of a short spindle was taken to indicate initiation of ring assembly and ring contraction was deemed to have initiated when the ring diameter started to reduce and the time for ring contraction was the duration between initiation of ring contraction and the complete disassembly of the ring.

4. In the text describing Fig 4A, "...displayed a significant reduction in F-actin length", it would be helpful to write upon addition of Adf1. Also, in the quantification of this experiment, μm is missing on the label to the graph y-axis and statistics is missing.

RESPONSE: This has been done as suggested.

August 19, 2019

RE: JCB Manuscript #201809089RR

Prof. Mohan K Balasubramanian
University of Warwick
Warwick Medical School The University of Warwick
Coventry CV4 7AL
United Kingdom

Dear Prof. Balasubramanian:

Thank you for submitting your revised manuscript entitled "Phospho-regulation of tropomyosin is crucial for actin cable turnover and division site placement". The reviewers have now assessed your revised paper and they both recommend acceptance so we would be happy to publish your paper in JCB pending final revisions necessary to meet our formatting guidelines (see details below).

****Please be sure to address the final remaining concern of reviewer #1 and provide a rebuttal in your cover letter.****

A. MANUSCRIPT ORGANIZATION AND FORMATTING:

Full guidelines are available on our Instructions for Authors page, <http://jcb.rupress.org/submission-guidelines#revised>. ****Submission of a paper that does not conform to JCB guidelines will delay the acceptance of your manuscript.****

- 1) Text limits: Character count for Reports is < 20,000, not including spaces. Count includes title page, abstract, introduction, results, discussion, acknowledgments, and figure legends. Count does not include materials and methods, references, tables, or supplemental legends. You are currently below this limit but please bear it in mind when revising.
- 2) Figure formatting: Scale bars must be present on all microscopy images, including inset magnifications. Molecular weight or nucleic acid size markers must be included on all gel electrophoresis.
- 3) Statistical analysis: Error bars on graphic representations of numerical data must be clearly described in the figure legend. The number of independent data points (n) represented in a graph must be indicated in the legend. Statistical methods should be explained in full in the materials and methods. For figures presenting pooled data the statistical measure should be defined in the figure legends. Please also be sure to indicate the statistical tests used in each of your experiments (both in the figure legend itself and in a separate methods section) as well as the parameters of the test (for example, if you ran a t-test, please indicate if it was one- or two-sided, etc.). Also, since you used parametric tests in your study (e.g. t-tests, ANOVA, etc.), you should have first determined whether the data was normally distributed before selecting that test. In the stats section of the

methods, please indicate how you tested for normality. If you did not test for normality, you must state something to the effect that "Data distribution was assumed to be normal but this was not formally tested."

4) Materials and methods: Should be comprehensive and not simply reference a previous publication for details on how an experiment was performed. Please provide full descriptions (at least in brief) in the text for readers who may not have access to referenced manuscripts.

5) Please be sure to provide the sequences for all of your primers/oligos and RNAi constructs in the materials and methods. You must also indicate in the methods the source, species, and catalog numbers (where appropriate) for all of your antibodies.

6) Microscope image acquisition: The following information must be provided about the acquisition and processing of images:

a. Make and model of microscope

b. Type, magnification, and numerical aperture of the objective lenses

c. Temperature

d. imaging medium

e. Fluorochromes

f. Camera make and model

g. Acquisition software

h. Any software used for image processing subsequent to data acquisition. Please include details and types of operations involved (e.g., type of deconvolution, 3D reconstitutions, surface or volume rendering, gamma adjustments, etc.).

7) References: There is no limit to the number of references cited in a manuscript. References should be cited parenthetically in the text by author and year of publication. Abbreviate the names of journals according to PubMed.

8) Supplemental materials: There are strict limits on the allowable amount of supplemental data. Reports may have up to 3 supplemental figures. At the moment, you are at this limit but please bear it in mind when revising.

Please also note that tables, like figures, should be provided as individual, editable files. A summary of all supplemental material should appear at the end of the Materials and methods section.

9) eTOC summary: A ~40-50 word summary that describes the context and significance of the findings for a general readership should be included on the title page. The statement should be written in the present tense and refer to the work in the third person.

10) Conflict of interest statement: JCB requires inclusion of a statement in the acknowledgements regarding competing financial interests. If no competing financial interests exist, please include the following statement: "The authors declare no competing financial interests." If competing interests are declared, please follow your statement of these competing interests with the following statement: "The authors declare no further competing financial interests."

11) ORCID IDs: ORCID IDs are unique identifiers allowing researchers to create a record of their various scholarly contributions in a single place. At resubmission of your final files, please consider providing an ORCID ID for as many contributing authors as possible.

B. FINAL FILES:

-- High-resolution figure and video files: See our detailed guidelines for preparing your production-ready images, <http://jcb.rupress.org/fig-vid-guidelines>.

Thank you for this interesting contribution, we look forward to publishing your paper in Journal of Cell Biology.

Sincerely,

Daniel Lew, PhD
Monitoring Editor
JCB

Tim Spencer, PhD
Deputy Editor
Journal of Cell Biology

Reviewer #1 (Comments to the Authors (Required)):

The authors have done a nice job of addressing every reviewer comment. In particular, they have greatly strengthened the evidence for direct phosphorylation of Cdc8 by Pom1, which was a critical weakness in the initial version. I still feel that the actin cable staining in Figure 1F-G is not very clear,

but I also do not feel that the paper hinges on this result, particularly given the very strong phenotypic results related to cytokinesis. I have one very minor comment for improvement - the y-axes for graphs in Figure 2H and 2J go down instead of up, which was confusing at first glance. The authors might flip these graphs to help readers easily understand the result. In general, this paper provides a nice mechanistic insight into control of cytokinesis by post-translational modification. I expect that it will be appreciated by the cell biology community.

Reviewer #2 (Comments to the Authors (Required)):

The authors have appropriately answered my comments with additional experiments. This is a very nice story.